# Using xPIRT to Record Pharmacy Interventions: An Observational, Cross-Sectional and Retrospective Study

**DOI:** 10.3390/healthcare10122450

**Published:** 2022-12-05

**Authors:** Rafael Baptista, Mary Williams, Jayne Price

**Affiliations:** 1Medicines Management, Powys Teaching Health Board, Hafren Ward, Bronllys Hospital, Brecon LD3 0LU, UK; 2School of Pharmacy & Pharmaceutical Sciences, Cardiff University, King Edward VII Avenue, Cardiff CF10 3NB, UK

**Keywords:** community hospital, patient safety, pharmacy intervention, prescribing error

## Abstract

Medication errors and omissions can potentially cause harm, prolong a hospital stay, lead to co-morbidities and even death. Pharmacy interventions (PI) ensure that these errors are identified and addressed, leading to improved patient safety and prescriber practice. Particularly in community hospitals, many only having general practitioners and not specialist doctors in their medical teams, PIs assume a strategic role. The PIs recorded throughout 8 months (between November 2021 and June 2022) in the community hospital wards in Powys, Wales, UK, using xPIRT (Pharmacy Intervention Recording Tool), a new pharmacy intervention record toolkit, were subjected to a retrospective analysis. The data were organised by location, drug, severity, acceptance, cost avoidance and intervention type. Significant prescribing errors were identified, which can potentially be different from those recorded in acute settings. Our results also informed on the need for integrated electronic prescribing systems paired with a PI recording tool to address effectively prescribing inaccuracies. Overall, this study was able to identify pharmacy teams as key to improve patient safety and care while contributing to significant cost-savings, through the recording of PI using xPIRT.

## 1. Introduction

The interventions of pharmacy professionals are considered valuable inputs towards optimised patient care and safety, by rationalising prescriptions and reducing and preventing medication errors [1,2]. On the wards, pharmacy teams can improve patient’s education and medication adherence. For the management of pharmacy teams, the record of interventions can inform on the service planning, training needs and clinical governance [1,3,4,5,6,7,8].

Particularly in community hospitals, many only having general practitioners (GPs) and not specialist doctors in their medical teams, pharmacy interventions (PIs) assume an important role in informing medics on their prescription practice. While PIs have been widely reported in the literature, for example in acute hospital settings or community pharmacies, this has not been seen for community hospitals [9]. Healthcare in community hospitals finds its focus on bridging the gap between an inpatient stay in acute hospitals and community care, concentrating on rehabilitation, safe discharge planning and medication optimisation and compliance [10].

Different strategies have been used to record PIs. Perhaps the most impactful so far is the development of Act-IP in 2006, with the introduction of the CLEO (Clinical, Economic and Organisational) scale in Act-IP v2, which has been measuring the clinical, economic and organisational impacts of Pis in several hospitals in France [11,12,13,14]. A recording tool should be easy to use as well as complex enough to capture all the dimensions of the interventions. While it is known that the development of new tools should consider the optimal theoretical, pragmatic and psychometric projections, many lack standardised documentation systems or robust information technology tools [14].

The present paper aims to present and demonstrate the use of a novel pharmacy intervention recording toolkit, xPIRT (Pharmacy Intervention Recording Tool), which was used to tackle the main drawbacks of the current PI recording platforms. This study also aims to describe the interventions recorded on xPIRT during a period of 8 months in all the community hospitals at Powys Teaching Health Board, Powys, Wales.

xPIRT is a user-friendly, live and free digital toolkit that allows for the recording of a PI. It also automatically organises data into useful infographics, using several Microsoft Office 365 Apps, such as Microsoft Forms, Microsoft Automator and Microsoft Power BI. xPIRT also considers the CLEO measuring model and integrates PIs into measurable impacts based on the conclusions of the EQUIP (Enhancing the Quality of User Involved Care Planning) study [15] and the cost-avoidance ScHAAR (School of Health and Related Research, the University of Sheffield) model [16,17].

## 2. Materials and Methods

### 2.1. Study Design and Setting

A retrospective, cross-sectional, observational study was performed for the PIs recorded using xPIRT in 8 community hospitals in Powys, with an average capacity for 151 patients and an occupancy rate of 92%. All the hospital inpatients between 1 November 2021 and 30 June 2022 were eligible. The patients’ demographic data were not recorded.

### 2.2. xPIRT and Recording Process

xPIRT is a digital toolkit that uses several functionalities of Microsoft Office 365. The data were recorded using a form on Microsoft Forms, particularly on the date, hospital, ward, contributor, medicines, type of intervention, outcome, timeline, intervention severity, acceptance and whether a Yellow Card or DATIX were reported. Only a brief description of each intervention was required, making xPIRT a user-friendly tool. An explanation for each topic can be seen in Appendix A. This Microsoft Forms is completed via laptop/desktop or smartphone by each member of staff on the pharmacy team when or after an intervention is completed on the hospital wards. Each submitted ‘Form’ corresponds to a single intervention. The easiness of use of this tool allows for minimal attention to be drawn from the patients’ care, but its broader impact, not only within the pharmacy team but also on the wards, it is yet to be known.

The error severity and calculated cost-avoidance were based on the EQUIP study, CLEO tool and ScHAAR cost-avoidance model (Appendix A) [11,15,16]. Internal validation was completed by five pharmacists of different seniorities. Each pharmacist independently and blindly evaluated the clinical impact of ten randomly selected interventions, followed by a team meeting to reach a consensus.

Data from the Microsoft Forms were organised in real-time from a list on Microsoft Lists using a Microsoft Power Automator function. This list was managed by one pharmacist for the standardisation and correction of slight inaccuracies such as typos. Data were automatically retrieved from Microsoft Lists to an interactive dashboard (xPIRT Dashboard) on Microsoft Power BI (Appendix A). Any data input on the Microsoft Forms were updated hourly, so the dashboard was constantly up to date. This was only possible using a Microsoft Power BI Premium account. The xPIRT Dashboard is shared and used within the team on Microsoft Teams.

### 2.3. Definition and Categorisation of Interventions

Various interventions were captured by the pharmacy team, including adverse drug reactions, the calculation of the dose given, the drug not given, form, route, timing, an incomplete or ambiguous drug chart and key notes on the drug administration that may have an impact on the clinical outcomes. Supply interventions were undertaken to avoid a delay in treatment or raise issues on incorrect storage. All PIs on prescribing were classified into 21 categories: allergy status incomplete, venous thromboembolism prophylaxis assessment incomplete, drug chart transcription incomplete, incorrect drug chart re-write or reconciliation, missing date or route, missing signature next to drug, missing mental health status, illegible handwriting, medicine duplication, dose/strength wrong or to be optimised, frequency/timing wrong or to be optimised, route is wrong or to be optimised, form is wrong or to be optimised, drug–drug interaction, drug–disease interaction, unnecessary drug therapy, initiation of therapy for untreated condition, therapeutic monitoring, medication review and/or stop dates, therapeutic switch for cost-effectiveness or formulary and therapeutic switch with clinical reasons.

### 2.4. Statistical Analysis

The descriptive analyses were conducted using Microsoft Excel, with the data extracted directly from Microsoft Lists. The ratio of PIs of a different severity per intervention type was organised into a heatmap, while the ratio of PIs of a different severity per the Anatomic Therapeutic Chemical (ATC) code was plotted into a circular graph.

## 3. Results

Pharmacy professionals (pharmacists and pharmacy technicians) were able to record and analyse their interventions. Each intervention inserted in the xPIRT took on average 2 min. Within 8 months, 735 interventions were recorded using xPIRT the the majority related to the prescribing practice (95.9%, *n* = 705), followed by a drug administration (1.9%, *n* = 14) and the drug supply (1.5%, *n* = 11). While the drug administration PIs were mainly aimed at advising nursing staff on the best practices for optimal drug therapy, the drug supply PIs avoided delays in the availability or distribution of drugs.

Overall, the PIs focused on 203 drugs (*n* = 720). The drugs mostly involved in the PIs were dalteparin (18.1%, *n* = 130), paracetamol (9.7%, *n* = 70) and omeprazole (3.3%, *n* = 24), while the therapeutic classes that were mostly involved in the PIs were group B ‘blood and blood forming organs’ (24.8%, *n* = 179), group A ‘alimentary tract and metabolism’ (18.7%, *n* = 135), group N ‘nervous system’ (17.7%, *n* = 128) and group J ‘anti-infectives for systemic use’ (8.6%, *n* = 62). The main sub-classes were B10A ‘antithrombotic agents’ (18.0%, *n* = 130), N02 ‘analgesics’ (10.0%, *n* = 72), A02 ‘drugs for acid related disorders’ (6.2%, *n* = 45) and B03 ‘antianemic preparations’ (6.1%, *n* = 44) (Table 1).

In terms of the severity, most of the PIs recorded were significant (76.9%, *n* = 555), followed by minor (18.4%, *n* = 133) and serious (4.7%, *n* = 34). No potentially lethal PIs were recorded. Drugs belonging to groups B, A and N contributed for 62.7% (*n* = 348) and 58.6% (*n* = 78) of significant and minor PIs, respectively (Figure 1). The drugs belonging to ATC groups B and C ‘cardiovascular system’ represented more than half the total number of severe PIs (52.9%, *n* = 18). The drug mostly involved in serious interventions was dalteparin, from group B, the low molecular weight heparin (LMWH) of choice for PTHB hospitals for venous thromboembolism (VTE) prophylaxis. Most serious PIs referred to the VTE risk assessment not being undertaken. This means that dalteparin was not prescribed for patients to whom that drug would be indicated and appropriate. Group C’s serious interventions focused on different drugs (digoxin, isosorbide mononitrate, glyceryl trinitrate, midodrine, nicorandil, furosemide, amlodipine, lisinopril and ramipril) and were involved in several types of interventions such as ‘unnecessary therapy’, ‘therapeutic monitoring, ‘missing drug (chart re-write or reconciliation)’, ‘drug-disease interaction’, ‘dose/strength wrong or to be optimised’ and the ‘initiation of therapy for untreated condition’. Group N’s serious interventions focused not only on opioid analgesics, such as morphine sulphate and oxycodone, but also on antiparkinsonians (co-careldopa) and anti-vertigo preparations (betahistine and cinnarizine) mainly due to the ‘dose/strength (wrong or to be optimised)’, ‘therapeutic switch—clinically justified’ and ‘drug-disease interaction’. The drugs involved in serious PI focused on the interventions that needed to be addressed at that time as patients were in acute distress (Appendix A).

The types of PI on the prescribing practice mainly focused on the dose/strength (wrong or to be optimised) (17.7%, *n* = 124), missing drug during the chart re-write or reconciliation (15.4%, *n* = 108), unnecessary drug therapy/deprescribing (9.5%, *n* = 67), and missing signatures on the drug chart (7.3%, *n* = 51). In terms of the severity, the types of PIs with the most serious interventions done were the ‘initiation of therapy for untreated condition’ (28.6%, *n* = 6), ‘therapeutic switch—clinically justified’ (28.6%, *n* = 4) and ‘drug-disease interaction’ (16.3%, *n* = 5) (Figure 2). The PIs involved in minor errors focused on missing signatures on prescription charts and illegible handwriting.

Overall, 81.1% (*n* = 596) of the PIs were accepted, while 0.4% (*n* = 3) were not accepted by the prescribers. For a total of 136 PIs (18.5%), the acceptance outcome was not recorded either because this was not applicable or the pharmacy staff were not aware of the outcomes upon the PI record.

Based on the ScHAAR model, the PIs recorded during 8 months in the community hospitals at PTHB were associated with a cost-avoidance between GBP 59,992 and GBP 133,862, meaning that the annual savings would achieve a figure above GBP 200,000.

## 4. Discussion

This study describes the interventions recorded by pharmacy professionals, pharmacists and pharmacy technicians in the community hospitals in Powys, Wales. Powys Teaching Health Board does not have a district general hospital but a network of community hospitals, local primary care teams, visiting consultants and specialists, social care and voluntary organisations that work together to provide a wide range of health services. Generally, Powys’ community hospitals provide general medicine, rehabilitation and palliative care services, with a few of them offering specialist mental health and stroke and neuro rehabilitation. Urgent care, such as minor injury units, are not included in this study; accident and emergency departments are not located in Powys. Registered pharmacy professionals are responsible and accountable for ensuring that medicines, medical gases and vaccines are prescribed, supplied, stores, prepared, disposed of and administered correctly and effectively. Pharmacists and pharmacy technicians provide specialist knowledge, medicines management and clinical expertise and work collaboratively with other healthcare staff and patients.

It is particularly important to describe the PIs undertaken in such hospitals as this has been under-reported in the literature. To the best of our knowledge, only one article has described the PIs of community hospitals [18]. Commonly, the medic teams of community hospitals are composed of GPs and, contrarily to acute hospitals, specialist doctors are not straightforwardly available. Thus, it is expected that the PIs could contain important information that would suggest the optimisation of specific medicines targeted at GPs in community hospitals. Studies conducted in acute hospitals may be of limited relevance to the prescribers that only practice in community hospitals.

Although previous results showed that the rate of prescribing errors in community hospitals is similar to that of acute settings [18], a drug-specific analysis has shown additional information. In community hospitals, the medications more associated with errors were antithrombotics, while acute settings have reported analgesics (opioid and non-opioid) and antibacterial drugs [18,19,20,21]. Particularly to our study, a large proportion of the errors were linked not only to the dose adjustment of LMWH and non-vitamin K antagonist oral anticoagulants (NOACs), but also the lack of a formal risk assessment for the use/indication and consequent omission of LMWH for the medical prophylaxis of venous thromboembolism. Nonetheless, our report also identified analgesics and antimicrobials as being two of the drug classes where errors were likely to occur. A distinctive high number of interventions has also been observed for ATC A02 ‘drugs for acid related disorders’, differently to what has been reported in the literature [19,20,21]. The PIs focused on over-prescribed histamine H2 receptor antagonists to replace the protein pump inhibitors (PPIs) in patients with electrolyte deregulation, and by the general PPIs over-prescribing occasionally with no clear indication.

Drug omissions (in drug chart re-writes or on admission), dose changes (wrong dose or to be optimised) and de-prescribing were the three main type of PIs found. The ‘drug chart’ refers to the combined prescribing and administration charts broadly used in the UK. Although studies have grouped PIs in different types, our results are partly in accordance with the EQUIP study. Interestingly, de-prescribing has not been the focus of the majority of the PIs made in acute settings [15]. Community hospitals may offer the time, care and opportunity required for the safer taper or discontinuation of medications, that at times can be challenging in acute settings. Regarding the second PI type, we argue that new methods, such as electronic prescribing, may effectively tackle drug omissions. With the advent of electronic prescribing software, it is expected that newer reports would identify less PIs on omitted drugs due to more efficient prescribing tools and a swift transfer of care between hospital settings.

The reported acceptance rate of PIs by prescribers varied widely. The high acceptance rate of this study is in line with others, who reported that the reason for high acceptance rates is due to the integration of pharmacy professionals in the multidisciplinary team [19]. Moreover, information technologies are known to influence positively and negatively the outcome of PIs. While good information systems allow pharmacy staff to raise drug-specific issues in an effective way, the lack of face-to-face clinical decisions leads to overlooked PIs. Ideally, electronic prescribing systems would have combined a tool to record and analyse PIs, similarly to xPIRT. xPIRT allowed the live integration of data recorded into an interactive dashboard with useful infographics. These data were widely available for the ward pharmacy and wider teams as it used free Microsoft Office 365 Apps.

Few considerations should be contemplated when interpreting the results of this study: (1) the record of PIs was not mandatory, so the results here presented may be reflecting only a portion of all the PIs conducted on the wards; (2) the PIs were recorded by a range of pharmacists and pharmacy technicians of varied seniorities; (3) the total number of patients receiving medication subjected to a PI was not recorded, so the intervention rate was not calculated; (4) while xPIRT was internally validated, an external validation may be required to fully authenticate the use of the tool in other healthcare organisations; (5) the calculation of cost-avoidance generated by PIs use the gold-standard ScHAAR model, which may require updating to match the most up-to-date figures for interventions on different error severities or intervention levels; and (6) finally, although each intervention inserted in the xPIRT took on average 2 min, the impact on the attention drawn from the direct patient care by the pharmacy team is yet to be assessed. It is expected that the success of this tool would be related to the pharmacy team’s motivation to record PIs, in order to generate usable/representative data.

It should be noted that this observational study does not compare each PIs like for like. There were no consideration of the intrinsic variances related with PIs done in different wards, covered by different medics, from different professionals of distinctive seniorities and job descriptions, under different workload pressures. Although the findings of this study mirror well the PIs of Powys community hospitals, due to the limited data available in the literature, we were not able to compare with the community hospitals of other health organisations.

## 5. Conclusions

This study demonstrated the use of a new pharmacy recording tool, xPIRT, and highlighted the interventions recorded during a period of 8 months in all the community hospitals at Powys Teaching Health Board, Powys, Wales.

The use of this tool allowed us to obtain representative data, particularly around prescribing medications in community hospitals. This is one of the few reports of PIs in community hospitals and, to the best of our knowledge, the first done in Wales. Contrary to what is perceived, inpatients in community hospitals are not focus of less PIs than in acute settings. The significant number of PIs recorded with xPIRT provided an opportunity to assess the priorities in prescribing, service planning and clinical learning.

Due to the use of xPIRT in Powys, new clinical guidelines are being produced; the medicines management/pharmacy team is now involved in several regular harm and mortality or clinical governance meetings to present and explain the prescribing data. More awareness has been generated for the value of pharmacists and pharmacy technicians.

This study clearly identifies pharmacy teams as key in generating important data that could inform on patient safety and care, and cost-savings opportunities. If undetected, many of the PIs recorded could have led to significant and serious harm.

## Figures and Tables

**Figure 1 healthcare-10-02450-f001:**
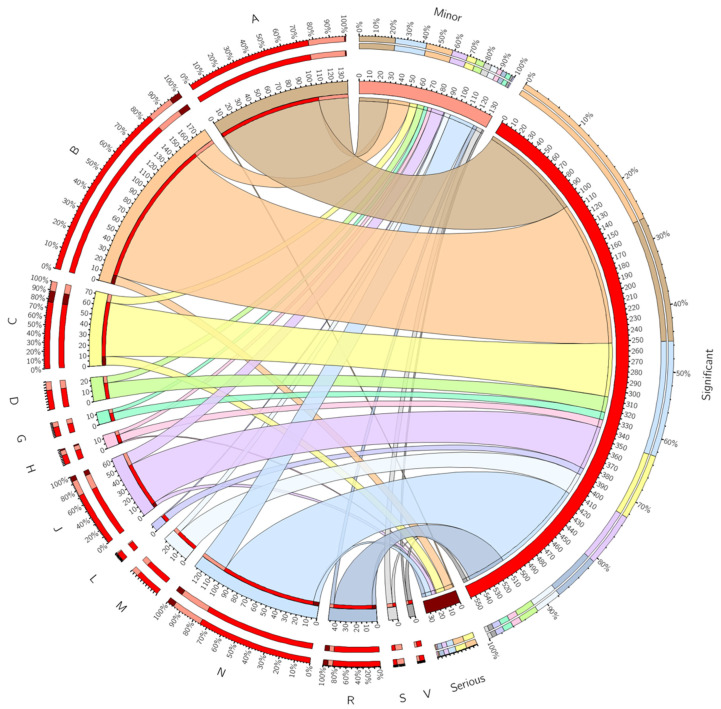
Relationship between the number and percentage of drugs focus of pharmacy interventions (PI), organized by Anatomic Therapeutic Chemical (ATC) code, and their severity (minor, significant and serious), via Circos. On the left of the plot, each slice represents a different ATC, while on the right, each segment represents PIs recorded organised by severity. Each ribbon or link represent the total number of PIs that belong to a particular ATC code and PI severity. ATC code: A—alimentary tract and metabolism, B—blood and blood forming organs, C—cardiovascular system, D—dermatologicals, G—genito urinary system and sex hormones, H—systemic hormonal preparations excl. sex hormones and insulins, J—anti-infectives for systemic use, L—antineoplastic and immunomodulating agents, M—musculo-skeletal system, N—nervous system, R—respiratory system, S—sensory organs, V—various.

**Figure 2 healthcare-10-02450-f002:**
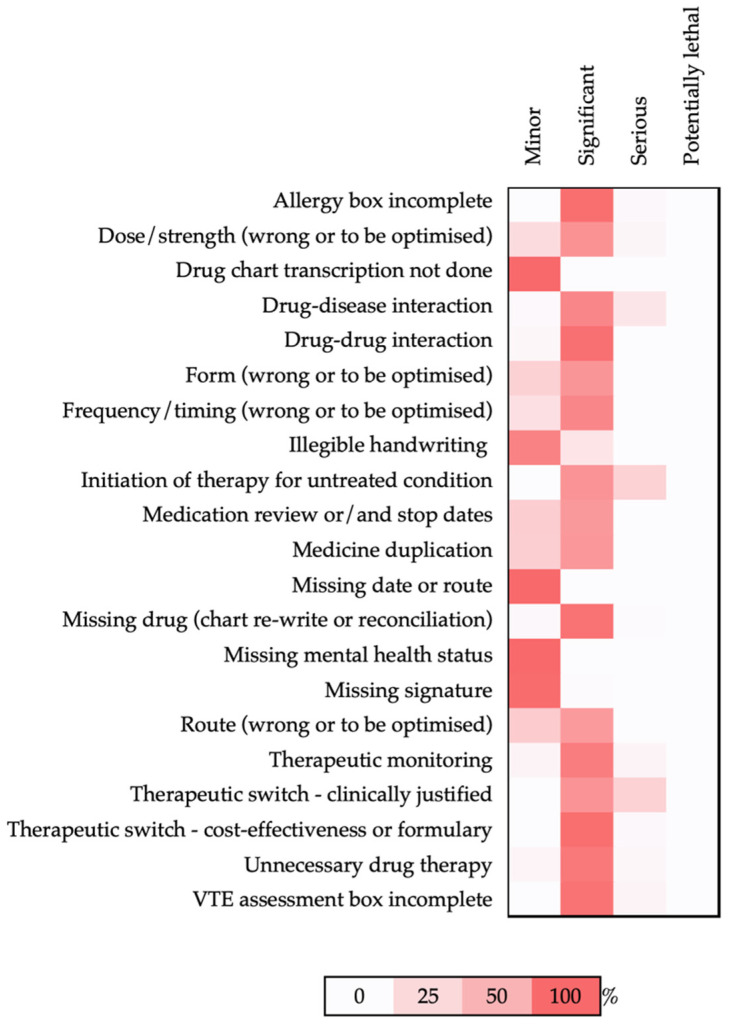
Heatmap of percentages of different types of PIs per severity. Scale bar represents different intensities of red according to the percentages.

**Table 1 healthcare-10-02450-t001:** Number of pharmacy interventions (PI) identified (*N*) by Anatomic Therapeutic Chemical (ATC) code and main drugs affected. Drugs with less than two interventions were not described in the table.

Anatomic Therapeutic Chemical (ATC) Code	*N*	Main Drugs Affected
A02 Drugs for acid related disorders	45	Omeprazole, famotidine
A03 Drugs for functional gastrointestinal disorders	7	Metoclopramide
A05 Bile and liver therapy	1	Ursodeoxycholic acid
A06 Drugs for constipation	18	Macrogols, senna
A07 Antidiarrheals, intestinal anti-inflammatory/anti-infective agents	7	Nystatin
A10 Drugs used in diabetes	12	Insulins, metformin, gliclazide
A11 Vitamins	26	Colecalciferol, vitamin B complex
A12 Mineral supplements	19	Calcium carbonate and colecalciferol, magnesium aspartate
B01A Antithrombotic agents	130	Dalteparin, apixaban
B03 Antianemic preparations	44	Folic acid, ferrous fumarate
B05 Blood substitutes and perfusion solutions	4	
C01 Cardiac therapy	13	Isosorbide mononitrate, glyceryl trinitrate, digoxin
C02 Antihypertensives	3	Doxazosin
C03 Diuretics	11	Furosemide, indapaminde
C05 Vasoprotectives	1	Diltiazem
C07 Beta blocking agents	6	Bisoprolol, sotalol
C08 Calcium channel blockers	8	Amlodipine, felodipine
C09 Agents acting on the renin-angiotensin system	17	Ramipril, losartan
C10 Lipid modifying agents	13	Atorvastatin, simvastatin
D Dermatologicals	23	Clotrimazole, miconazole
G Genito-urinary system and sex hormones	12	Mirabegron, conjugated oestrogens (equine)
H02 Corticosteroids for systemic use	7	Prednisolone, fludrocortisone
H03 Thyroid therapy	4	Levothyroxine, carbimazole
J Anti-infectives for systemic use	62	Doxycycline, nitrofurantoin
L01B Antimetabolites	4	Methotrexate
L02 Endocrine therapy	4	Leuprorelin
M Musculo-skeletal system	28	Alendronic acid, ibuprofen
N02 Analgesics	72	Paracetamol
N03 Antiepileptics	3	
N04 Anti-Parkinsons drugs	3	Co-careldopa, procyclidine
N05A Antipsychotics	10	Risperidone, olanzapine, haloperidol
N05B Anxiolytics	5	Lorazepam, diazepam
N05C Hypnotics and sedatives	5	
N06A Antidepressants	21	Mirtazapine, citalopram, sertraline
N06D Anti-dementia drugs	1	Memantine
N07B Drugs used in addictive disorders	6	Nicotine
N07C Antivertigo preparations	2	Cinnarizine, betahistine
R01A Decongestants and other nasal preparations for topical use	1	Mometasone
R03 Drugs for obstructive airway diseases	34	Salbutamol, beclomethasone dipropionate with formoterol
R06 Antihistamines for systemic use	12	Cyclizine, loratadine, promethazine
S Sensory organs	9	Bimatoprost, chloramphenicol
V06 General Nutrients	6	Nutritional supplements, enteral feed

## Data Availability

Not applicable.

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
