# Peer review of "Using xPIRT to Record Pharmacy Interventions: An Observational, Cross-Sectional and Retrospective Study"

_healthcare, 2022, doi:10.3390/healthcare10122450_

Round 1

Reviewer 1 Report

Thank you for the interesting paper. There are some issues that need to be addressed:

in the introduction -  find more supporting literature for the first two paragraphs -  there have been many interventions, including  pharmacists lead one, with no actual benefit in the terms of length of stay, readmission etc. Please be more careful in these statements and support them with stronger references. 

What is the purpose of the present study? to present the tool or to describe most common interventions or both? in the present form, none of these aims come across strong, which is unfortunate since the tool and the scope of interventions performed and tracked seems important.

Specifically - the setting must be described in more detail, including organisation and responsibilities of pharmacy care there. This widely differs and must be described. 

The way PI were classified needs to be described in more detail, not just mentioned by references. 

Process of recording PI is unclear. I understand this was self-reported by each pharmacy care person performing it? This is mentioned in limitations, but briefly and should be reflected  more in the methods and limitations need to be broadened.

Figure 1 looks amazingly pretty but needs a bit of an explanation on how to read it in the caption.

Please be more clear on the implications of this study. Why it is important? 

Author Response

Reviewer 1

in the introduction -  find more supporting literature for the first two paragraphs -  there have been many interventions, including  pharmacists lead one, with no actual benefit in the terms of length of stay, readmission etc. Please be more careful in these statements and support them with stronger references. 

Thank you for this review. These two paragraphs were re-written, now just with references to the role of pharmacy teams in “rationalising prescriptions, reducing and preventing medication errors”, “improve patient’s education and medication adherence”. No other statements are done, which could have opened a broader discussion, as this is not the aim of this paper.

What is the purpose of the present study? to present the tool or to describe most common interventions or both? in the present form, none of these aims come across strong, which is unfortunate since the tool and the scope of interventions performed and tracked seems important.

Thank you for your important remarks. The introduction was re-written and the aims of the study are now explained: “The present paper aims to present and demonstrate the use of a novel pharmacy intervention recording toolkit, xPIRT (Pharmacy Intervention Recording Tool), which was used to tackle the main drawbacks of the current PI recording platforms. This study also aims to describe the interventions recorded on xPIRT during a period of 8 months in all community hospitals at Powys Teaching Health Board, Powys, Wales.” This has been also highlighted in the conclusion section of this paper.

Specifically - the setting must be described in more detail, including organisation and responsibilities of pharmacy care there. This widely differs and must be described. 

Thank you for your comment and suggestion, this information is quite relevant. We added a paragraph to the discussion: “This study describes the interventions recorded by pharmacy professionals, pharmacists and pharmacy technicians, in the community hospitals in Powys, Wales. Powys Teaching Health Board does not have a district general hospital but a network of community hospitals, local primary care teams, visiting consultants and specialists, social care, voluntary organisations that work together to provide a wide range of health services. Generally, Powys’ community hospitals provide general medicine, rehabilitation and palliative care services, with a few of them offering specialist mental health and stroke and neuro rehabilitation. Urgent care, such as minor injury units, are not included in this study; accident & emergency departments are not located in Powys. Registered pharmacy professionals are responsible and accountable for ensuring that medicines, medical gases and vaccines are prescribed, supplied, stores, prepared, disposed and administered correctly and effectively. Pharmacists and pharmacy technicians provide specialist knowledge, medicines management and clinical expertise and work collaboratively with other healthcare staff and patients.”

The way PI were classified needs to be described in more detail, not just mentioned by references. 

Thank you for this remark. We believe the classification of interventions is fully explained (and exemplified) in the Supplementary Table 2. This table was internally validated by five pharmacists of different seniorities. Each pharmacist independently and blindly evaluated the clinical impact of ten randomly selected interventions, following by a team meeting to reach consensus. We hope this aligns well with the reviewer’s suggestion.

Process of recording PI is unclear. I understand this was self-reported by each pharmacy care person performing it? This is mentioned in limitations, but briefly and should be reflected  more in the methods and limitations need to be broadened.

Thank you for the suggestion. The following paragraphs were re-written:

(1)

“xPIRT is a digital toolkit that uses several functionalities of Microsoft Office 365. Data was recorded using a form on Microsoft Forms, particularly on the date, hospital, ward, contributor, medicines, type of intervention, outcome, timeline, intervention severity, acceptance, whether Yellow Card or DATIX reported. Only a brief description of each intervention was required, making xPIRT a user-friendly tool. An explanation for each topic can be seen in Supplementary Table 1. This Microsoft Forms is completed via laptop/desktop or smartphone by each pharmacy team staff when or after an intervention is completed on the hospital wards. Each submitted ‘Form’ corresponds to a single intervention. The easiness of use of this tool allows minimal attention to be drawn from patients care, but its broader impact, not only within the pharmacy team but also on the wards, it is yet to be known.”

(2)

“Few considerations should be contemplated when interpreting the results of this study: (1) the record of PIs was not mandatory, so the results here presented may be reflecting only a portion of all the PIs done on the wards; (2) PIs were recorded by a range of pharmacists and pharmacy technicians of varied seniorities; (3) the total number of patients with medication subjected to a PI was not recorded, so the intervention rate was not calculated; (4) while xPIRT was internally validated, an external validation may be required to fully authenticate the use of the tool in other healthcare organisations; (5) finally, the calculation of cost-avoidance generated by PIs use the gold-standard ScHAAR model, which may require updating to match the most up-to-date figures for interventions on different error severities or intervention levels; (6) although each intervention inserted in the xPIRT took on average 2 minutes, the impact on attention drawn from direct patient care by the pharmacy team is yet to be assessed. It is expected that the success of this tool would be related with the pharmacy team’s motivation to record PIs, in order to generate usable/representative data.”

Figure 1 looks amazingly pretty but needs a bit of an explanation on how to read it in the caption.

Thank you for your comment, this is really appreciated. The legend of the figure has been re-written: “Figure 1. Relationship between the number and percentage of drugs focus of pharmacy interventions (PI), organized by Anatomic Therapeutic Chemical (ATC) code, and their severity (minor, significant and serious), via Circos. On the left of the plot, each slice represents a different ATC, while on the right, each segment represents PIs recorded organised by severity. Each ribbon or link represent the total number of PIs that belong to a particular ATC code and PI severity. ATC code: A – alimentary tract and metabolism, B – blood and blood forming organs, C – cardiovascular system, D – dermatologicals, G – genito urinary system and sex hormones, H – systemic hormonal preparations excl. sex hormones and insulins, J – antiinfectives for systemic use, L – antineoplastic and immunomodulating agents, M – musculo-skeletal system, N – nervous system, R – respiratory system, S – sensory organs, V – various.”

Please be more clear on the implications of this study. Why it is important? 

Thank you for your suggestion for improvement. The implications are now organised and described in each paragraph in the conclusion: a) the fact that this new intervention recording tool can be used on the wards; b) the new intervention tool allowed representative data collection in community hospitals, letting us to compare this study to others done previously (for example, prescribing interventions in acute settings vs community settings); c) intervention tool generated data that is changing clinical practice in Powys (for example, new clinical guidelines are being produced and pharmacy team is not part of clinical governance meetings in order to feedback and educate prescribers); d) establish pharmacy team as key in generating important data that could inform on patient safety, care and cost-savings opportunities.

The conclusion was re-written: “This study demonstrated the use of a new pharmacy recording tool, xPIRT, and highlighted the interventions recorded during a period of 8 months in all community hospitals at Powys Teaching Health Board, Powys, Wales.

The use of this tool allowed to obtain representative data particularly around prescribing in community hospitals. This is one of the few reports of PIs in community hospitals and, to the best of our knowledge, the first done in Wales. Contrary to what is perceived, inpatients in community hospitals are not focus of less requirement for PIs than in acute settings. The significant number of PIs recorded with xPIRT provided an opportunity to assess priorities in prescribing, service planning and clinical learning.

Due to the use of xPIRT in Powys, new clinical guidelines are being produced, the medicines management/pharmacy team is now involved in several regular harm and mortality or clinical governance meetings to present and explain prescribing data. More awareness has been generated for the value of pharmacists and pharmacy technicians.

This study clearly identifies pharmacy teams as key in generating important data that could inform on patient safety and care, and cost-savings opportunities. If undetected, many PIs recorded could have led to significant and serious harm.”

Reviewer 2 Report

Very interesting paper

Author Response

Many thanks for your comments.

Reviewer 3 Report

Why patients' demographics data wasn't recorded?

There is no specific information about statistical analysis section. When it comes to statistics, was  the heat map only used? Please supplement and clarify. 

Figure one is very unclear - can you show this data in another way?

Please explain if those filling out the database have received prior training and how the database was constructed.

The literature should be made consistent and unified.

Conclusion is sparse. Reference could be made to meetings of drug management teams currently underway and referred to by the authors.

Author Response

Why patients' demographics data wasn't recorded?

Thank you for this review and this particular question. Patient’s demographics wasn’t recorded in this study or with xPIRT as this would potentially add barriers to the use of the tool (for example, patient data confidentiality), as we are using open source Microsoft apps, and also because the prescribing inaccuracies identified by the recording tool are not patient-specific, but drug- and ward-specific.

There is no specific information about statistical analysis section. When it comes to statistics, was  the heat map only used? Please supplement and clarify. 

Thank you for the raised questions. There is no relevant statistical data treatment in this study, as described in our results (descriptive analysis). Data was only organised and described by groups. The heatmap and the circular graph were the only ones used in this paper.

Figure one is very unclear - can you show this data in another way?

Thank you for the raised question and suggestion. We would argue that Figure 1 is able to organise a large number of PIs in a different way, not losing any information that would be easily missed with the use of a 3-entry table. Reviewer 1 referred to this figure as “amazingly pretty”. We have re-written the legend of the figure, so the reader can understand how to ‘read’ this figure easily: “Figure 1. Relationship between the number and percentage of drugs focus of pharmacy interventions (PI), organized by Anatomic Therapeutic Chemical (ATC) code, and their severity (minor, significant and serious), via Circos. On the left of the plot, each slice represents a different ATC, while on the right, each segment represents PIs recorded organised by severity. Each ribbon or link represent the total number of PIs that belong to a particular ATC code and PI severity. ATC code: A – alimentary tract and metabolism, B – blood and blood forming organs, C – cardiovascular system, D – dermatologicals, G – genito urinary system and sex hormones, H – systemic hormonal preparations excl. sex hormones and insulins, J – antiinfectives for systemic use, L – antineoplastic and immunomodulating agents, M – musculo-skeletal system, N – nervous system, R – respiratory system, S – sensory organs, V – various.”

Please explain if those filling out the database have received prior training and how the database was constructed.

Thank you for your important remarks. No formal training was given prior to the use of this tool, as this was thought to be unnecessary. Only an informal introduction to the tool was provided in a pharmacy team meeting. Regarding the construction of the tool, the following paragraph was re-written: “xPIRT is a digital toolkit that uses several functionalities of Microsoft Office 365. Data was recorded using a form on Microsoft Forms, particularly on the date, hospital, ward, contributor, medicines, type of intervention, outcome, timeline, intervention severity, acceptance, whether Yellow Card or DATIX reported. Only a brief description of each intervention was required, making xPIRT a user-friendly tool. An explanation for each topic can be seen in Supplementary Table 1. This Microsoft Forms is completed via laptop/desktop or smartphone by each pharmacy team staff when or after an intervention is completed on the hospital wards.”

The literature should be made consistent and unified.

Thank you for your comment. References are now formatted accordingly.

Conclusion is sparse. Reference could be made to meetings of drug management teams currently underway and referred to by the authors.

Thank you for your comment. Conclusion has been re-written: “This study demonstrated the use of a new pharmacy recording tool, xPIRT, and highlighted the interventions recorded during a period of 8 months in all community hospitals at Powys Teaching Health Board, Powys, Wales.

The use of this tool allowed to obtain representative data particularly around prescribing in community hospitals. This is one of the few reports of PIs in community hospitals and, to the best of our knowledge, the first done in Wales. Contrary to what is perceived, inpatients in community hospitals are not focus of less requirement for PIs than in acute settings. The significant number of PIs recorded with xPIRT provided an opportunity to assess priorities in prescribing, service planning and clinical learning.

Due to the use of xPIRT in Powys, new clinical guidelines are being produced, the medicines management/pharmacy team is now involved in several regular harm and mortality or clinical governance meetings to present and explain prescribing data. More awareness has been generated for the value of pharmacists and pharmacy technicians.

This study clearly identifies pharmacy teams as key in generating important data that could inform on patient safety and care, and cost-savings opportunities. If undetected, many PIs recorded could have led to significant and serious harm.”

Reviewer 4 Report

The article “Using xPIRT to record pharmacy interventions: an observational, cross-sectional and retrospective study” is focused on an interesting topic, overall, the article is well written but I have the following comments/suggestions,

1. Kindly define PIRT in the abstract.

2. In supplementary Table 2 I am unable to follow how the cost for the intervention was calculated.

3. The procedure for a pharmacy intervention in Supplementary Figure 1 is somewhat vague, it needs further explanation in the text.

4. Can the authors comment on how the ethical approval was not applicable to this study?

Author Response

The article “Using xPIRT to record pharmacy interventions: an observational, cross-sectional and retrospective study” is focused on an interesting topic, overall, the article is well written but I have the following comments/suggestions,

  1. Kindly define PIRT in the abstract.

Thank you for this review and this particular comment. Abstract has been corrected accordingly.

  1. In supplementary Table 2 I am unable to follow how the cost for the intervention was calculated.

Thank you for this question. The cost calculated for each intervention has been based on ScHAAR cost-avoidance model, reference 17 on this study. This has been described in the Materials and Methods section.

  1. The procedure for a pharmacy intervention in Supplementary Figure 1 is somewhat vague, it needs further explanation in the text.

Thank you for this comment, shared with all the other reviewers. Materials and methods section has been re-written: “xPIRT is a digital toolkit that uses several functionalities of Microsoft Office 365. Data was recorded using a form on Microsoft Forms, particularly on the date, hospital, ward, contributor, medicines, type of intervention, outcome, timeline, intervention severity, acceptance, whether Yellow Card or DATIX reported. Only a brief description of each intervention was required, making xPIRT a user-friendly tool. An explanation for each topic can be seen in Supplementary Table 1. This Microsoft Forms is completed via laptop/desktop or smartphone by each pharmacy team staff when or after an intervention is completed on the hospital wards. Each submitted ‘Form’ corresponds to a single intervention.” (…) “Data from the Microsoft Forms was organised in real-time from a list on Microsoft Lists using a Microsoft Power Automator function. This list was managed by one pharmacist for standardisation and correction of slight inaccuracies such as typos. Data was automatically retrieved from Microsoft Lists to an interactive dashboard (xPIRT Dashboard) on Microsoft Power BI (Supplementary Figure 1).”

  1. Can the authors comment on how the ethical approval was not applicable to this study?

Many thanks for this question. Ethical approval was not applicable to this non-interventional service evaluation study as no personal identifiable data has been recorded. This ‘no applicability’ status has been obtained by the Powys Teaching Health Board Research and Development Department.

Round 2

Reviewer 1 Report

All comments were addressed

Reviewer 4 Report

The authors have addressed all of my comments in their revised submission.